# Design of Intelligent Management Platform for Industry–Education Cooperation of Vocational Education by Data Mining

**Min Wu [1], Xinxin Hao [1], Yang Lv [2] and Zihan Hu [3],***

1 Department of Public Affairs Management and Public Policy, School of Public Administration, Sichuan University, Chengdu 610065, China; wu_min@scu.edu.cn (M.W.); haoxinxin@stu.scu.edu.cn (X.H.)
2 Department of Preschool Education, College of Teachers, Chengdu University, Chengdu 610106, China; lvyang@cdu.edu.cn
3 Faculty of Business, University of Prince Edward Island, Charlottetown, PE C1A 4P3, Canada
* Correspondence: zhu3@upei.ca

**Abstract:** Data are playing an increasingly important role in the development of industry–education cooperation strategies in vocational education and training. The objective of this study was to promote the comprehensive progress of an industry–education cooperation system and improve the effect of the application of big data technology in this system. First, we designed of a big data technology application in an intelligent management platform system for industry–education cooperation. Second, we analyzed the synthetical design of the system. Finally, we optimized and designed a support vector machine (SVM) data mining (DM) algorithm model based on big data, and evaluated the model. The results revealed that the designed algorithm model provides outstanding advantages compared with similar algorithm models. In general, the highest average computation time of the designed SVM algorithm model is about 95 ms. The overall average calculation time linearly decreases around 200 iterations and tends to be stable, and the lowest overall average computation time is about 20 ms. In the DM process, the highest accuracy rate of the model is about 97%, and the lowest is about 92%. The DM accuracy rate is always stable as the number of iterations of the model continues to increase. The designed model slowly increases the occupancy rate of the system in the process of increasing computing time. At about 60 min, the system occupancy rate of the model tends to be stable, and the highest is maintained at about 23%. This study not only provides technical support for the optimization of DM algorithms with big data technology, but also contributes to the integrated development of industry–education cooperation systems.

**Keywords:** industry–education cooperation; big data technology; the SVM data mining algorithm; intelligent management platform

## 1. Introduction

Vocational education and training (VET) is a "safety net" for labor force employment in society because it better meets the needs of the labor market compared with general education, thereby improving the status of vocational-education students and reducing the possibility of future unemployment of VET graduates [1]. Graduates with work experience can more smoothly enter the workforce, especially during the COVID-19 pandemic [2]. Therefore, in order to improve the construction of VET platforms and optimize the current vocational education effect, in this work, we used data mining (DM) algorithms to build a reasonable VET platform.

The dual system VET in German is internationally recognized as performing very well [3–5]. It is characterized by the joint participation of companies, schools and colleges, government, and industry associations. Students receive a formal education at a vocational school, along with practical training driven by the needs of the company, and then

directly practice in the company as an apprentice [6,7]. China has been trying to adopt the German VET system; however, due to the long-term monolithic management of the Chinese government, the original characteristic of multiple partners being involved in German VET system has not been transferred [8]. The Chinese VET currently has some limitations. School/college–enterprise cooperation is inefficient, and human, material, and information resources are not provided quickly during the cooperation process [9,10]. It is difficult to transmit the needs of enterprises to schools in time. The skills that students learn from formal and practical training courses in vocational schools and colleges are not what companies actually need [11,12].

China has started to develop an industry–education cooperation strategy in order to solve these VET problems. The core of the aim of industry–education cooperation is to promote the connection between the education and talent chains of vocational schools and colleges and the industrial and innovation chains of enterprises, which means that the people trained at vocational schools and colleges have the work skills that companies really need [13]. Industry–education integration can deepen partners' involvement in VET and enhance the efficiency of information sharing [14]. The in-depth cooperation between industry and education is the main direction of modern VET reform [15].

The increasing employment of new technologies can help people in various aspects of life [16]. Advances in technology have provided important technical support for deepening the degree of industry–university cooperation in VET. In the context of the swift growth of big data technology, the deep integration of big data and education has become a trend; with the rapid progress of education informatization, big data technology will provide vital technical support for higher-education reform. Higher vocational schools and colleges can rely on big data technology to deepen cooperation with companies and innovate talent training methods [17]. Therefore, using network technology to establish a network platform for remedial teaching is an effective teaching tool and the best way to achieve the sharing of information and resources [18]. By establishing an intelligent management platform for industry–education cooperation, it can open up channels for data resource sharing among partners and improve the development speed of VET [19]. Although this technology is not mature enough, many studies have provided technical support for it. Intelligent service platforms can realize the interconnection of information between main and substations, and improve the efficiency of information resource sharing [20]. The 3D virtual display on the web side and VR technology on the mobile side enable a more comprehensive display of teaching results [21]. Information technology and data mining (DM) technology are changing people's lives, and information processing has become stronger. Various types of information research are becoming increasingly important [22].

Although big data technologies can help VET partners more effectively share data resources, little research has focused on the design of data resource development platforms for industry–education cooperation [23]. The existing research has mainly focused on sharing online VET resources [24], VET credit certification [25], and VET quality evaluation [26] using data mining. Therefore, the main contribution of our study is the design of an intelligent management platform for industry–university integration based on DM technology. We optimized a support vector machine (SVM)-DM algorithm model for the data acquisition of the intelligent management platform for industry–education cooperation. This study not only provides technical support for the compositive optimization of big data technology, but also contributes to the rapid advancement of strategies for industry–education cooperation.

Section 2 introduces the related research theory and methods. The design of DM technology in big data technology is discussed, and the application and design of the VET intelligent management platform are studied. Section 3 presents the setup of the study data. Section 4 details the performance evaluation of the vocational education intelligence platform with DM. Section 5 discusses the behavioral logic of VET partners and the positive impact of the intelligent management platform on their behavior. Conclusions are drawn and future work is discussed in Section 6. The contribution of this study is that it provides

a basic DM algorithm for the construction of VET, and makes the construction of the VET platform more active through DM.

## 2. Research Theory and Methods

### 2.1. Big Data Technology

The term big data, also known as mega data, means that the amount of data is so huge that it cannot be processed by ordinary data software tools. Big data must be collected, managed, processed, and organized within a reasonable time to provide reference information to help the business decision making [27]. It refers to the technology of collecting data from various industries and quickly obtaining valuable information through these data. The emergence of big data has provided a critical technical foundation for the rapid collection of information, and has an important impact on social development. It has many characteristics, for instance, huge data volume, diverse data types, fast processing speed, and low value density [28].

Based on the various characteristics of big data, its role is also integrated. First, its processing and analysis are becoming the characteristics of the new generation of information technology fusion applications. Technologies such as mobile Internet, the Internet of Things (IoT), digital homes, social networks, and e-commerce are some forms of application of the new generation of information technology in this era; these technologies are all serving big data technology, constantly generating a variety data [29]. Additionally, cloud computing technology provides a platform for parallel computing and storage of these massive and diverse big data. These platforms feed the final results back to the above-mentioned applications through a series of operations on data from different sources, creating huge economic and production value for social development. Therefore, if big data technology is interpreted as an industry, this industry profits through processing data, thereby improving the value of data use and realizing the continuous increase in data value [30]. Second, big data are the main force driving the continuous change in the information industry and the continuous accumulation of information. New services, products, technologies, and formats for big data technology are constantly emerging. Moreover, big data technology has played a huge role in promoting the development of hardware and integrated equipment; that is, it is significantly influencing the chip manufacturing and output storage industries. It has motivated the development and mass production of integrated data storage processors, and its use will become a key factor in improving the core competitiveness of society. That is, big data technology will become the key technology supporting various industries, and will become the main force driving the development of various industries. Finally, the methods and means of scientific research in the era of big data technology are undergoing major changes. This era will produce a variety of methods to improve the efficiency and value of scientific research [27]. The overall design of big data technology platforms is displayed in Figure 1.

As shown in Figure 1, as the current main pillar of industry, big data technology can be used to conduct experimental, innovative research and can increase the value of research. Furthermore, its technical means are various, involving visual analysis, DM, predictive analysis, semantic engines, and data management [31]. There are many trends in its development, including the recycling of data: big data have become a vital competitive resource in various industries in current society, and have become a new strategic point of competition. From this, for enterprises to develop in current society, they must formulate strategies and technologies related to big data in advance to seize the market opportunities created by big data [32]. Big data technology needs to be combined with cloud computing technology, meaning that its application and development are inseparable from cloud processing. Cloud computing technology provides the basic equipment for data processing, and is one of the main platforms for generating big data. Since the beginning of the 21st century, big data technology has been linked with cloud computing technology, so, in the future, the relationship between the two will be even closer. Additionally, a series of new scientific computing technologies, such as the IoT and mobile Internet, will

provide technical support for the development of big data technology, so that it can exert its more comprehensive advantages and promote the overall expansion of society [29]. The breakthroughs in scientific theory have led to big data technology creating a new era of scientific and technological reform. What has followed is the rapid development and application of related technologies such as DM, machine learning (ML) and artificial intelligence (AI), which are changing the status of data in the social industry, and leading to their own related theoretical knowledge construction, achieving breakthroughs in scientific computing technology through big data technology. The establishment of data science and data alliances have shown that data leakage incidents through big data technology will significantly increase in the next few years. Therefore, continuous optimization and reform are needed to enhance the main performance of big data technology [33]. The degree of compounding of the data ecosystem is strengthened, that is, big data technology is not a single scientific technology, but a synthetically applied scientific computing technology that integrates multiple technologies [34]. Nowadays, it has begun to take shape in social development. Therefore, various constructions must be designed to strengthen its performance and promote its application in social industries. Helping social development is the main future development direction of big data [35]. On account of their many advantages and broad development prospects in the future, the research platforms constructed with big data technology have great value.

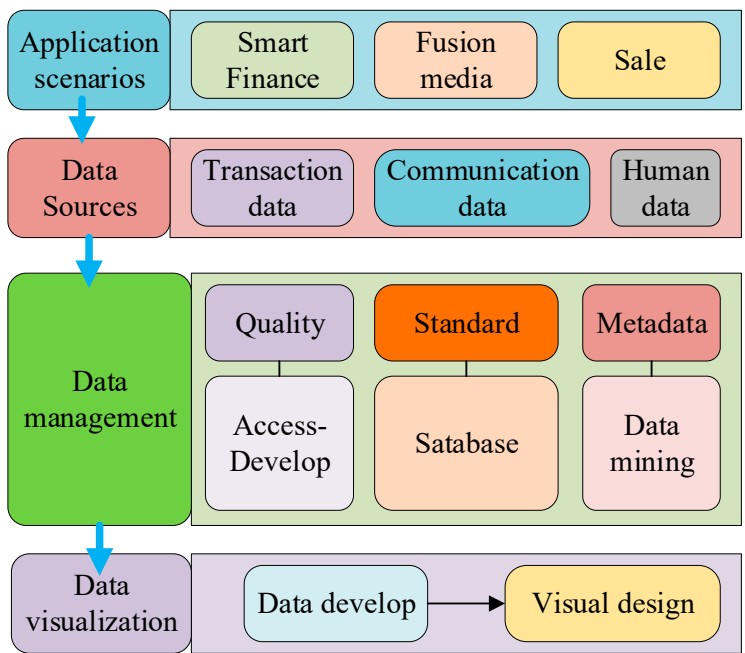

**Figure 1.** Overall design of big data technology platforms.

## 2.2. Industry–Education Cooperation of Vocational Education and Training

Vocational education and training allow an educated person to acquire the vocational knowledge, skills, and work ethics required for a certain occupation. From a macro perspective, industry–education cooperation means the coordinated development of VET and industrial systems. From a micro perspective, industry–education cooperation means that vocational schools and colleges offer courses based on the needs of companies, and teach students working knowledge and skills to improve the quality of training. [36]. VET is also a complex social system, comprising the government, higher vocational colleges, industrial enterprises, students, and other subjects. VET is externally affected by policy factors and economic factors, and internally by various elements such as educational concepts, school resources, and enterprise needs. It has a certain unique characteristics. However, the social views on industry–education cooperation are not unified: there are two mainstream views [37]. One view is that industry–education cooperation is school/college–

enterprise cooperation. This view advocates that school/college–enterprise cooperation and industry–education cooperation are the products of different stages of economic and social development. Although the expression and denotation are different, the connotation and essence are the same. The second view is that industry–education cooperation is not school/college–enterprise cooperation. This view argues that the connotation and essence of school–enterprise cooperation are not the same as industry–education cooperation, and that industry–education cooperation is an advanced stage of school–enterprise cooperation. In short, from a macro perspective, industry–education cooperation is the integration of industry and education, and from a micro perspective, it is the integration of production and teaching [38]. Therefore, industry–education cooperation has three basic characteristics. The first is diversified subjects. The management involves the government, and operation involves entities such as vocational education schools, production industries, and enterprises [39]. The second is dynamic evolution. The fundamental purpose is to promote the adaptation of VET to social and economic development. The third is cross-system cooperation, which is the integration of the VET and industrial systems. In conclusion, our basic aim in this study was to link VET and industrial systems and build an information management platform to enable industry–education cooperation. We aimed to help the VET system obtain more real industrial system information, and promote the rapid growth of VET systems. We also provide more comprehensive VET information for the industrial system, and integrate teaching elements into its production process [40]. Our design approach for industry–education cooperation is shown in Figure 2.

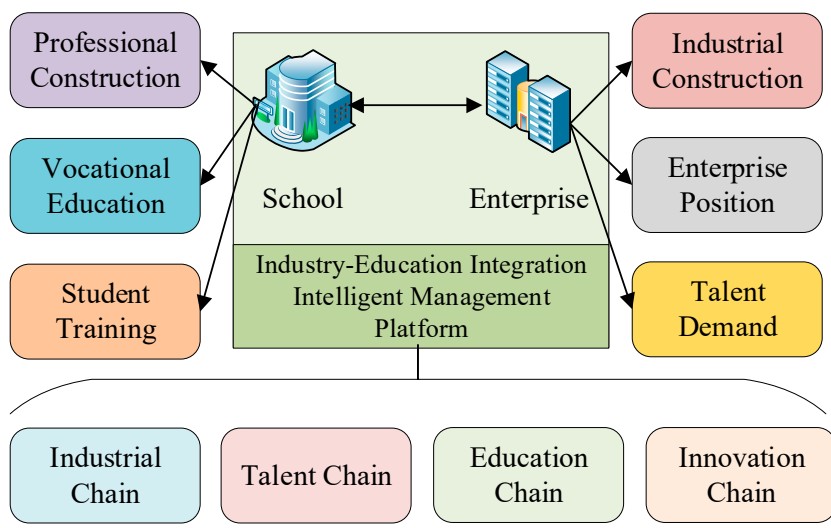

**Figure 2.** The design approach for industry–education cooperation.

In Figure 2, industry–education cooperation, as a product of the close connection between current VET and social-industrial systems, provides vital support for promoting the two [41]. We mainly designed an intelligent management platform through the DM technology of big data, providing technical support for industry–education cooperation systems and promoting its comprehensive progress.

### 2.3. Mining Technology under Big Data

DM must first be used to preprocess the data. This necessitates some additional processing of the acquired data before the big data technology processing [42]. The main purpose of DM is to provide more accurate data that meets the requirements and improves the efficiency and accuracy of information processing by big data technology. This operation is one of the important information processing aspects of big data technology. There are various methods for data preprocessing, including data integration, cleaning, selection, etc. [43]. With the support of these data preprocessing technologies, the synthetical quality

of DM technology in practical applications is greatly improved, the time required for actual DM is reduced, and the practicability of actual data acquisition is improved.

Data cleaning is the integrated improvement in incomplete, noisy, and inconsistent data in the real world. Namely, through data filling, noise reduction, and prediction methods, the data are cleaned to improve their integrity and practicability [44]. The requirements of the results of data cleaning comprise the standardization of data format, ensuring no data are abnormal, removing erroneous data, and filling f blank data. Data integration is the unified processing of multiple data sources in massive data sets to improve the effectiveness of data storage [45]. If the form of the original data is not suitable for the information processing algorithm, data transformation is necessary. Data integration involves combining data from multiple data sources into a consistent store. Data selection involves filtering and selecting the data to be used. Big data technology mainly includes the K-nearest neighbors (KNN), naïve Bayes, k-means, apriori, and PageRank algorithms, and SVM. In this study, we principally designed and optimized the SVM algorithm to enable its application in an intelligent management platform for industry–education cooperation, to improve the efficiency of the platform DM, and promote its development [46].

The VSM algorithm is an example of a class of machine algorithms that perform generalized linear classification of data by supervised learning, and its decision boundary is the maximum margin hyperplane that solves the learning samples. Therefore, its main core is a linear classification algorithm, and its input data and learning objectives are:

$$X = \{x_1, \ldots, x_N\} \tag{1}$$

$$y = \{y_1, \ldots, y_N\} \tag{2}$$

where $X$ represents the input data, and $y$ is the learning objective. If there is a hyperplane as the decision boundary in the feature space where the input data are located, the learning targets are separated into positive and negative classes. If the point-to-plane distance of any sample is greater than or equal to one, then the calculation for the decision boundary and the point-to-plane distance is as follows:

$$w^\top X + b = 0 \tag{3}$$

$$y_i\left(w^\top X_i + b\right) \geq 1 \tag{4}$$

where $w^\top$ denotes the normal vector of the hyperplane, and $b$ indicates the intercept of the hyperplane. Then, the hyperplane is used as the interval boundary to discriminate the sample classification algorithm, as exhibited in Equations (5) and (6):

$$w^\mathrm{T} X_i + b \geq +1, \Rightarrow y_i = +1 \tag{5}$$

$$w^\mathrm{T} X_i + b \leq -1, \Rightarrow y_i = -1 \tag{6}$$

The implication is that all samples above the upper interval boundary belong to the positive class, and those below the lower interval boundary belong to the negative class. The calculation of the loss function of the SVM algorithm is expressed in Equation (7):

$$L(p) = \begin{cases} 0 & p < 0 \\ 1 & p \geq 0 \end{cases} \tag{7}$$

where $p$ is probability. The 0-1 loss function is not a continuous function, so is not conducive to the solution of optimization problems. So, the usual choice is to construct a proxy loss. The loss function used by SVM after optimization is the hinge loss function, and its calculation is demonstrated in Equation (8):

$$L(p) = \max(0, \ 1 - p) \tag{8}$$

Consistency studies on surrogate losses showed that when the surrogate loss is a continuous convex function and is an upper bound of the 0-1 loss function at any value, the result of solving the surrogate loss minimization is also the solution of the 0-1 loss minimization. The hinge loss function satisfies the above conditions. According to statistical learning theory, a classifier generates risk when it learns on and is applied to new data. The risk can be divided into empirical and structural risk types. The calculation is as follows:

$$\epsilon = \sum_{i=1}^{N} L(p_i) = \sum_{i=1}^{N} L[f(\mathbf{X}_i, \mathbf{w}), y_i] \tag{9}$$

$$\Omega(f) = \| \mathbf{w} \|^p \tag{10}$$

where $f$ refers to the classifier, $N$ is the total number of samples, and $i$ denotes the calculation sample. The equations show that empirical risk is defined by the loss function, and structural risk is defined by the classifier parameter matrix. Then, a classifier determines its model through two sublines, and the calculation of the model is indicated in Equations (11) and (12):

$$\mathcal{L} = \| \mathbf{w} \|^p + C \sum_{i=1}^{N} L[f(\mathbf{X}_i, \mathbf{w}), y_i] \tag{11}$$

$$\mathbf{w} = \arg\min_{w} \mathcal{L} \tag{12}$$

where $C$ is the regularization coefficient. The risk of the above model has been minimized. The improved algorithm for skewed data is written in Equation (13)

$$C_{+1} N_{+1} = C_{-1} N_{-1} \tag{13}$$

where $+1$ and $-1$ represent positive and negative examples, respectively. The improved calculation of probability SVM is illustrated in Equations (14) and (15):

$$\hat{A}, \hat{B} = \arg\min_{A,B} \frac{1}{N} \sum_{i=1}^{N} (y_i + 1) \log(p_i) + (1 - y_i) \log(1 - p_i) \tag{14}$$

$$p_i = \text{sigmoid}\left[ \hat{A}\left( \mathbf{w}^\top \phi(\mathbf{X}_i) + b \right) + \hat{B} \right] \tag{15}$$

where $\hat{A}$ and $\hat{B}$ express the zoom and translation parameters, respectively. The solution of the model is as follows:

$$h(\boldsymbol{\alpha}, \beta) = -\sum_{i=1}^{N} \alpha_i + \frac{1}{2} \sum_{i=1}^{N} \sum_{j=1}^{N} (\alpha_i Q \alpha_j) + \sum_{i=1}^{N} I(-\alpha_i) + \sum_{i=1}^{N} I(\alpha_i - C) + \beta \sum_{i=1}^{N} \alpha_i y_i \tag{16}$$

$$I(x) = -\frac{1}{t} \log(-x), Q = y_i (\mathbf{X}_i)^\top (\mathbf{X}_j) y_j \tag{17}$$

where $I$ represents the logarithmic blocking function, which essentially uses a continuous function to approximate the inequality relationship in the constraints, and the meanings of the calculation parameters in all the above equations are the same. Based on the DM technology used with big data, the SVM algorithm was optimized, and the VET intelligent management platform was constructed. The algorithm comprehensively improved the efficiency and accuracy of data acquisition of the intelligent management platform, and provided more professional and accurate educational reference in-formation for the vocational education system. The platform can provide better professional information for the social industry system, promote the development of the social industry, and improve the efficiency of integrating teaching elements into its production process [47]. Figure 3 depicts the design method we used to improve the performance of the intelligent VET management platform by optimizing the SVM algorithm.

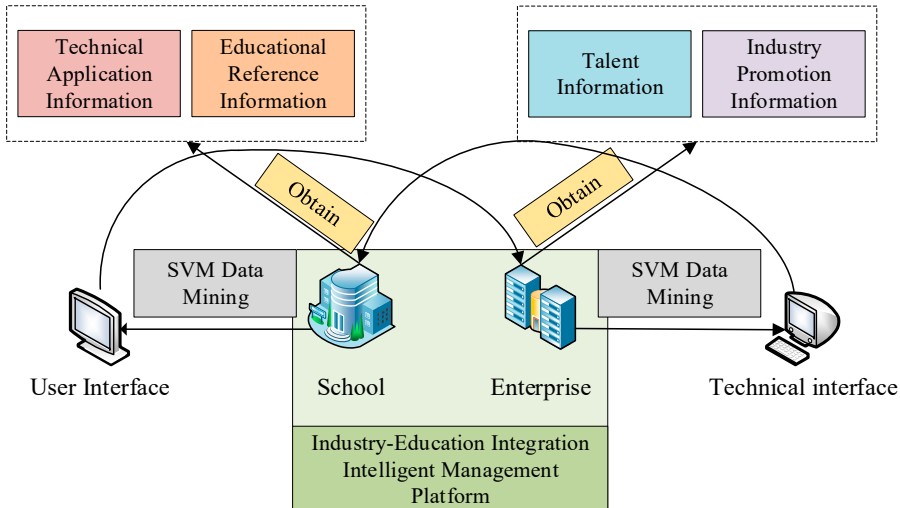

**Figure 3.** The design method of optimizing the SVM algorithm to improve the performance of an intelligent management platform enabling industry–education cooperation.

Figure 3 shows that optimizing the SVM algorithm enhanced the performance of the intelligent management platform of industry–education cooperation using big data technology. Through the overall application of two-way data, the development of social industry systems can be promoted, and the teaching efficiency of the VET system can be increased.

## 3. Setup of Research Data

We designed a comprehensive SVM-DM algorithm, optimized the VET intelligent management platform, increased the integration of the performance of the platform, and increased the accuracy of the information provided by the platform. When evaluating the designed technology, we used various data sets for training evaluation. The datasets used were as follows:

(1) The Abalone dataset, which contains a variety of factors, is a multivariate dataset. It comprises 4177 instances, and the attribute features have 8 attributes such as classifications, integers, and real numbers.

(2) The Adult dataset, which is the U.S. census dataset, is a multivariate dataset. It includes 48,842 instances, and the 14 attribute features include, e.g., classification and integer.

(3) The Covertype dataset, which is a multivariate dataset, as 581,012 instances, and the attribute features have 54 attributes, such as classifications, integers, and real numbers.

(4) The Nomao dataset is a collection of data about places from many sources. It is a univariate dataset that includes 34,465 instances, and the attributes include 54 real-number attributes.

The above datasets are all open datasets and can be obtained through the Kaggle data platform. Based on the above datasets, the designed SVM-DM technology was comprehensively evaluated to determine the optimization effect of big data technology on the VET management platform. We compared and evaluated the designed model with other models: (1) The K-nearest neighbor (KNN) model is a simple and practical classification algorithm that measures the distance between different samples, and then selects the nearest K neighbors according to the distance for classification. The core idea of the algorithm is that if most of the K-nearest samples in the feature space belong to a certain category, the sample also belongs to this category. The whole classification process includes three steps: calculating the distance to determine which neighbors are closest; finding the nearest neighbor, that is, selecting the K value; then making a decision and classifying. The KNN algorithm is a classification technique that is easy to understand

and implement, and performs well in many situations. It is particularly suitable for multimodal classes and objects with multiclass labels. (2) The Naïve Bayes algorithm model is a classification algorithm based on Bayes' theorem that can be used for exploratory and predictive modeling. The algorithm uses Bayesian techniques but does not take into account possible dependencies. (3) The k-means algorithm model is relatively simple, using the cluster to represent the cluster. It is easy to prove that the convergence of the k-means algorithm is equivalent to all the centroids no longer changing. Its advantages include its simplicity, speed, high efficiency, and scalability for large data sets, and the time complexity is nearly linear, which is suitable for mining large-scale datasets. (4) The apriori algorithm model is the first association rule mining algorithm and a classic algorithm. It uses the iterative layer-by-layer search method to find the relationships of item sets in the database to form rules. The process consists of connection (matrix-like operations) and pruning (removing the unnecessary intermediate results). The concept of item sets in this algorithm is a collection of items. A set containing k items is a k-item set. The frequency of occurrence of an item set is the number of transactions that contain the item set. If an item set satisfies the minimum support, it is called a frequent item set. (5) The PageRank algorithm model basically defines a random walk model on a directed graph, that is, a first-order Markov chain, which describes the behavior of walkers randomly visiting each node along with the directed graph. Under certain conditions, the probability of visiting each node in the limit case converges to a stationary distribution. At this time, the stationary probability value of each node is its PageRank value, which indicates the importance of the node. PageRank is defined recursively, and its calculation can be performed by an iterative algorithm.

## 4. Performance Evaluation of Vocational Education Intelligence Platform under DM

### 4.1. Performance Evaluation of DM

The SVM-DM algorithm model was used to optimize and design the DM technology for big data. The intelligent management platform enabling industry–education cooperation was built through optimized SVM-DM technology. The main objective was to improve the efficiency of integration between the social industry and VET systems, and to deepen the degree of integration between the two, thereby promoting the development of social industry systems while improving the effect of VET. The comparison of the performance evaluation results of the designed SVM algorithm model to those of the other models are illustrated in Figure 4.

In Figure 4, the comparison of the results obtained by the optimally designed SVM algorithm with those of the other algorithms shows that the designed algorithm model has outstanding advantages. In essence, the highest average calculation time of the designed SVM algorithm model was about 95 ms; the overall average calculation time linearly decreased and tended to be stable after about 200 iterations. The overall average calculation time was at least about 20 ms. The other models started to decline when the number of iterations was about 300, and the calculation time of the other models was at least about 30 ms. This showed that the model pair has obvious optimization in DM. Figure 5 compares the evaluation results of the accuracy of the DM in the optimally designed SVM algorithm model with those of the other models.

Figure 5 signifies that the optimally designed SVM algorithm model provides advantages over the other algorithms in terms of the accuracy of DM on different data sets. Overall, the accuracy of the model was around 97% at the highest and around 92% at the lowest, and the accuracy of DM was always stable as the number of iterations of the model continued to increase. The accuracy of DM in the other models was generally around 90%, and the stability of DM in the other models was relatively poor. It can be seen that the designed model also has advantages in the accuracy of DM.

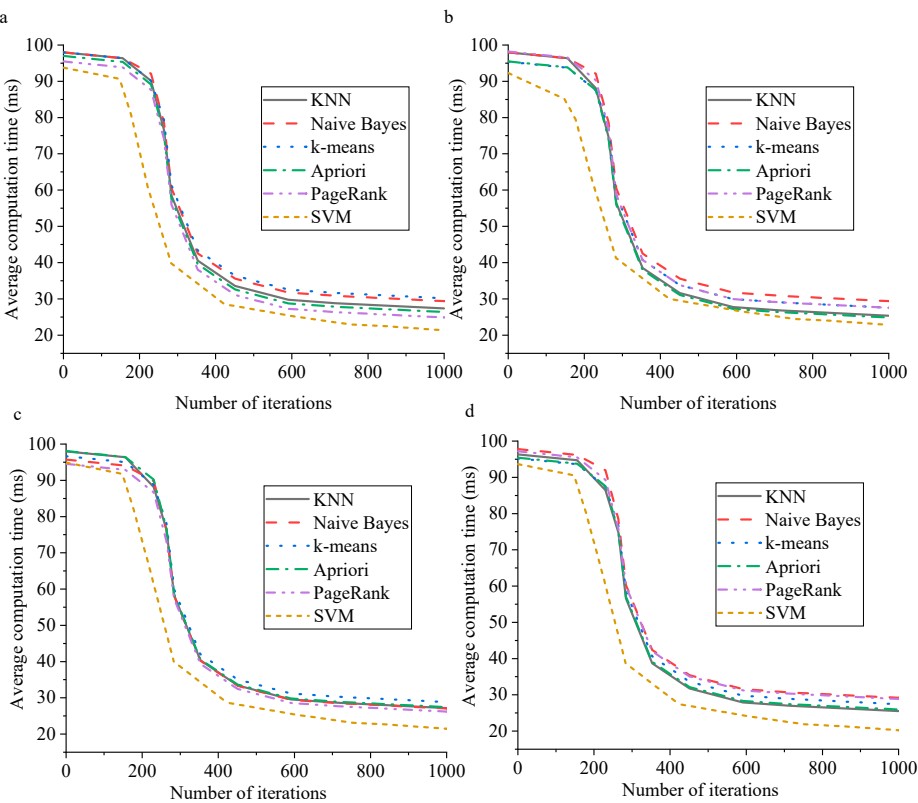

**Figure 4.** The comparative evaluation of the performance of the SVM algorithm model on the (**a**) Abalone, (**b**) Adult, (**c**) Covertype, and (**d**) Nomao datasets.

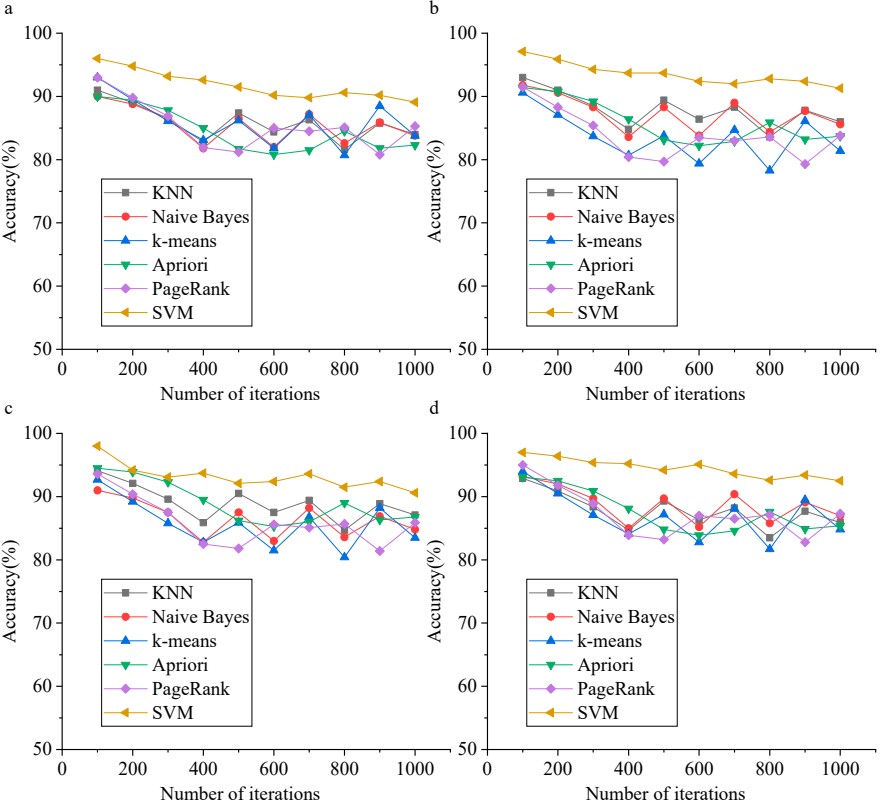

**Figure 5.** Comparative evaluation of the accuracy of DM in the SVM algorithm model on the (**a**) Abalone, (**b**) Adult, (**c**) Covertype, and (**d**) Nomao datasets.

### 4.2. Evaluation of the System Suitability of the SVM-DM Model

An algorithm should not only have strong computing properties, but also have better system applicability, so that the computing model can provide a good technical foundation for compositive applications. The designed SVM algorithm model not only provides a better computing model for the VET intelligent management platform, but also creates a better use environment for the system. Therefore, the occupancy rate of the system was evaluated through the SVM algorithm model in the calculation process to test the effect of using the SVM algorithm model. The designed models were evaluated in groups to test the comprehensive applicability of the models, divided into group-1, -2, and -3. The evaluation results of the system suitability of the optimally designed SVM algorithm model are shown in Figure 6.

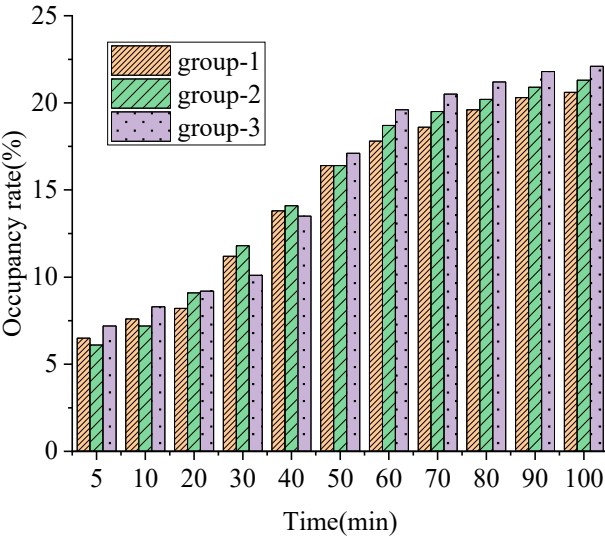

**Figure 6.** The evaluation of the system suitability of the SVM algorithm model.

Figure 6 shows to the evaluation results of the occupancy rate of the integrated system by the SVM algorithm model during the calculation process. The results indicated that the occupancy rate of the model to the system slowly increased as the computing time of the designed model increased. At about 60 min, the system occupancy rate of the model tended to be stable, and the system occupancy rate was maintained at about 23% at the highest. At the same time, the difference among the three groups was not large. Therefore, the optimized algorithm model can well-adapt to the VET intelligent management platform system.

## 5. Discussion

In vocational education systems, schools and colleges mainly provide the functions of personnel training, scientific research, social services, and cultural inheritance. Enterprises in the industry mainly pursue economic interests and assume social responsibilities. Cooperative education to meet different demands is essentially a mutual transaction of resources. In the case of information asymmetry, it may cause harm to those involved and increase transaction costs [48].

Based on this research result, we found that the highest average calculation time of the designed SVM algorithm model was about 95 ms, and the overall average calculation time linearly decreased and tended to be stable after about 200 iterations. The overall average calculation time was at least about 20 ms. The performance of the other models started to decline when the number of iterations was about 300, and the calculation time of other models was at least about 30 ms. Through the studies of the models in groups, we found that the occupancy rate of the model to the system slowly increased as the computing time of the designed model increased. A about 60 min, the system occupancy rate of

the model tended to be stable, and the system occupancy rate was maintained at about 23% at the highest. The difference among the three groups was not large. Therefore, we concluded that when enterprises participate in high-efficiency and low-cost VET, their motivation will be very high to participate. From the perspective of resource dependence, enterprises mainly participate in VET to obtain the external resources they need, especially the talent and technologies provided by schools and colleges. The higher the degree of dependence of the enterprise on the resources provided by the school and colleges, the lower the resource substitution and the deeper the enterprise's participation in vocational education. The time and energy, as resources invested by schools, colleges, and enterprises, further enhance this resource dependence and strengthen the school/college–enterprise partnership. According to the logic of the enterprise system, the force driving the deep participation of enterprises in VET is formed from the aspects of reducing the transaction cost of enterprises participating in vocational education and improving the supply capacity of schools [49].

In the intelligent platform we built for vocational education colleges, the participating partners include universities, enterprises, and social organizations. Although the needs of different types of partners are different, it is easier to maximize the utilization of resources through the platform and reduce the transaction costs of VET. Compared with the original VET model, the VET model we developed through the intelligent service platform provides the following advantages:

(1)    It is easier for the government to take the lead. The government can not only advocate for schools, colleges, and enterprises to join VET, but also better play its guiding role through policies, regulations, consultation, and supervision.

(2)    In this model, the scale of enterprise demand affects the scale of school and college supply; on the contrary, the scale of school and college supply also stimulates the expansion of the size of enterprise demand. The expansion of the scale of either the supply or demand side affects the other side, which can effectively stimulate the scale of participation of both schools and colleges and enterprises.

(3)    It can alleviate the information asymmetry among universities, governments, industry organizations, and enterprises, and facilitate the communicate of information and needs among all parties in a timely manner. Partners can independently decide the participation fee for the meeting establishing the relationship according to the supply and demand of the bilateral market, avoiding the restriction of one party providing special funds.

(4)    The platform model has no time limit and is more sustainable. Schools, colleges, and enterprises can formulate corresponding VET cooperation models in a timely manner according to each other's needs, thereby effectively increasing the frequency of exchanges and interactions between partners.

## 6. Conclusions

With the constant progress of society, social industry and vocational education have become important projects in society. The integration and development of the vocational education and the social industry systems through science and technology are innovative measures. This method is also important for promote the common development of the two systems. Based on this, in this study, we designed and applied big data technology to the current VET system. Consequently, we demonstrated the design basis of applications of big data technology in VET systems. Then, the design of the VET intelligent management platform was analyzed, and the optimization method was discussed. Finally, the SVM algorithm model was optimized and assessed, and the model wad comprehensively evaluated. First, the comparison between the optimally designed SVM algorithm and other algorithms demonstrated that the designed algorithm model has outstanding advantages. Generally, the highest average computation time of the designed SVM algorithm model is about 95 ms, the overall average computation time linearly decreases around 200 iterations, and tends to be stable. The overall average calculation time is at least about 20 ms, which means that the

model pair has obvious DM optimization. Second, in the DM process, the highest accuracy rate of the model is about 97%, and the lowest is about 92%. As the number of iterations of the model continues to increase, the DM accuracy rate is always stable. However, the DM accuracy of the other models we used for comparison is generally around 90%, and the stability of DM of other models is relatively poor. Third, the occupancy rate of the model in the designed model slowly increases with increasing computing time. The system occupancy rate of the model tends to be stable at about 60 min, and the system occupancy rate is maintained at about 23% at the highest. Although we provided a relatively complete model design and evaluation results, the research on the practical application of the model is not perfect. Hence, in the future, the research on the practical applications of the model could be strengthened to improve the comprehensive effect of using the model.

**Author Contributions:** M.W. led and designed the study, led the data analysis, and contributed to the acquisition of the financial support for the project leading to this publication. X.H. contributed to the study design, provided input on the data analysis, and wrote the first draft of the manuscript. Y.L. contributed to the study design, project administration, and reviewed and edited the first and final of the manuscript. Z.H. provided input into literature review, data analysis, and wrote the final draft of the manuscript. All authors have read and agreed to the published version of the manuscript.

**Funding:** The article is the research results of the Education Science Western Region Project entitled "The Behavior Logic and Realization Mechanism of Multiple Subjects' Synergetic Governance on Vocational Education" (Project No. XJA190284) funded by The National Social Science Fund of China.

**Institutional Review Board Statement:** Not applicable.

**Informed Consent Statement:** Not applicable.

**Data Availability Statement:** The data used to support the findings of this study are available from the corresponding author upon request.

**Conflicts of Interest:** The authors declare no conflict of interest.

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
