# Peer review of "Design of Intelligent Management Platform for Industry–Education Cooperation of Vocational Education by Data Mining"

_applsci, doi:10.3390/app12146836_

Round 1
Reviewer 1 Report
It's an interesting article, but the result is not consistent. The manuscript is poorly written, difficult to read. The methodology is not justified either. Thus, as it currently stands, I cannot recommend publishing this work in Applied Science.
Comments:
- Section 3. 'Setup of research data': Important information such as data provenance, essential metadata (download date, type, format, attribute name) are not specified. Those details must be defined in advance.
- The results also need to be explained in depth.
- Section ‘4.1. Performance evaluation of DM’
- Figure 4: the authors present the results of the comparison between the SVM algorithm designed with other algorithms (K Nearest Neighbours, Naive-Bayes, etc.). However, the other algorithms are not presented in details, just mentioned, and therefore it is in principle impossible to reproduce or validate the work used by the authors.
- Figure 5: It is not clear the comparative evaluation results of the accuracy between the algorithms. I suggest authors detail the performance of each algorithm shown in the graph.
- It should be clarified that this work only shows the perfamance of the SVM algorithm.
- Apart from that, I believe there is space to expand critical analysis on the limitations of the proposed methods, based on the limited amount of experimental data available.
- Figure 6 is completely confusing, there is no meaning of 'group-1, group-2, group-3', it must be added.
- Section Discussion must be supported by numerical results.
- Section Conclusion: This section needs to conclude what was done in the paper; their models, results, comparison of results, etc. The authors need to further validate their result.
- Finally, the novelty of the paper is not clear.
Reject - article has serious flaws, additional experiments needed, research not conducted correctly.
Author Response
To the reviewer: We were pleased to receive constructive comments from the reviewer. We appreciate a lot for all your efforts in reviewing this paper. We have carefully considered the suggestions from the reviewer and revised the manuscript. Please see the attachment.

Reviewer 2 Report
The goal of this research is to promote the overall progress of the industry-education cooperation system and to improve the application effect of big data technology in this system. First, the design of an application of big data technology in an intelligent management platform system for industry-education collaboration is discussed. The system's synthetical design is then examined. Finally, the big data-based Support Vector Machine (SVM) data mining (DM) algorithm model is optimized and designed, and the model is evaluated. However, I still have the following questions and suggestions.
In general, the paper is difficult to read and could be significantly improved. This paper's presentation is poor, and many major sections are difficult to understand. The main idea of this paper is difficult to grasp.
What are the paper's new challenges? What are the benefits of the method proposed in this paper over others?
The figures are distorted and should be corrected.
The authors should explain how the proposed Platform differs from other platforms. Some experiment details are vague.
The evaluation metrics have not been thoroughly discussed.
In fact, the authors only provide equations for each metric with no explanation.
Author Response
To the reviewer: We were pleased to receive constructive comments from the reviewer. We appreciate a lot for all your efforts in reviewing this paper. We have carefully considered the suggestions from the reviewer and modified the paper. Please see the attachment.

Round 2
Reviewer 1 Report
The manuscript has been revised. No further comments.